# Analysis of Design Method and Mechanical Properties of Plug-In Composite Shear Wall

**Xiang Sun [1], Peiyu Liu [2], Zhelong Jiang [2], Yuqing Yang [2], Zhe Wang [3] and Zaigen Mu [2,*]**

1   Shenzhen Wanchuang Architectural Design Consulting Co., Ltd., Shenzhen 518055, China
2   School of Civil and Resource Engineering, University of Science and Technology Beijing, Beijing 100083, China
3   China Institute of Building Standard Design & Research Co., Ltd., Beijing 100048, China
*   Correspondence: zgmu@ces.ustb.edu.cn

**Abstract:** Assembly buildings are an important direction for the future development of the construction field. They can be prefabricated in the factories and then assembled on-site, which significantly improves construction efficiency. The shear walls are the most important lateral force-resisting elements in building structures, and at this stage, there are more and more studies on the prefabricated shear wall. In this paper, a new composite shear wall structure is proposed. The composite shear wall is a part of a prefabricated building, which is prefabricated into a single shear wall unit in the factory. During the construction, the upper and lower prefabricated shear wall units are connected by the plug-in. The design methods of splicing connection are given for the design of this composite shear wall structure. Eleven composite wall models under different parameters are established by using the finite element method, especially the fine modeling of the upper and lower connection parts. Compared with the conventional composite shear wall model of the same dimensions, the mechanical behaviors of the two models are similar. In the simulation of cyclic loading, the new composite shear wall shows good ductility and energy dissipation capacity, and also meets the established requirements of building seismic performance. Therefore, it can be concluded that the new prefabricated composite shear walls have good development prospects and application values.

**Keywords:** finite element analysis; prefabricated building; composite shear wall; seismic performance; hysteretic curve

## 1. Introduction

At present, the shear wall structure is one of the most commonly used structural systems. The double steel plate-concrete composite shear wall is one of them. This kind of composite shear wall integrates the advantages of steel and concrete, which bear loads together and give full play to their respective advantages. The research of Link [1], Emori [2], Masahiko [3], and Eom [4] has proved that the double steel plate-concrete composite shear wall (DSCW) has excellent load-bearing capacity, ductility, and energy dissipation capacity. In recent years, Nie and Bu [5,6] have also conducted relevant studies on double steel plate-concrete composite walls.

China is in the stage of rapid development, while the construction industry also upholds the concept of sustainable development and tries to reach the goal of green building. The concept of green building has become a new hot spot in China's architecture field. The prefabricated shear wall structure can overcome the drawbacks of the long construction cycle, high cost, and heavy pollution of the cast-in-place shear wall. Therefore, to achieve longer-term development, the shear wall structure should adhere to innovation, adapt to the current development direction and channel great effort into the prefabricated shear wall which is reliable and economical.

The related research on prefabricated shear wall structures in China is still in the early stage, and there is also a lack of practical application in the engineering field. The research

on prefabricated shear walls still needs a lot of theoretical calculations and experimental research. To make the prefabricated shear walls reach the lateral stiffness of the conventional cast-in-place shear wall is the main question. The connections between its members have become the key research problem that needs to be solved in the whole structure. Scholars have conducted some relevant research on the connection mode. Wang et al. [7] carried out experimental research on the seismic performance of precast shear wall post-poured connection beam; Wang et al. [8] studied the connection design of assembled composite shear walls with concealed bracing and gave advice on relevant details for further research and design. Zhu [9] and Qian [10] conducted relevant experimental analysis on precast shear walls and designed the part of sleeve grouting connections. Wang et al. [11] established a model of precast shear walls with sleeve connections by finite element software and made a relevant simulation analysis. Xue [12] and Wang [13] conducted experimental studies on assembled shear walls connected by both bolts and casing pipes. Wei [14] and Sun [15] studied the structure of assembled shear walls connected by bolts through experiments and concluded that bolted connections can be used as a method for connections of prefabricated shear walls. Zhang [16] proposed a composite shear wall with a semi-rigid frame, where the precast concrete slabs are easily assembled in the form of segmental splices and connected to the frame by T-shaped damping connectors.

In recent years, new types of composite shear walls and their research have emerged. However, the issue of rapid installation for use in assembled buildings has not been addressed. In addition, the research related to assembled shear walls using segmental assembly and bolted connections is limited and the number of related literatures is small. This paper is based on such a background and proposes the assembled double steel plate-concrete combination shear wall through the innovative method of combining top and bottom assembling and bolting. By analyzing its mechanical properties and comparing it with the traditional combined shear wall model, the seismic performance of the new combined shear wall in practical engineering is investigated. The results of the study can provide references for future related research and engineering applications.

## 2. Design Method and Model Establishment

### 2.1. Composition of Composite Shear Wall

This new composite shear wall connected by bolts is convenient in joint and lifting construction with a high degree of assembly and other good characteristics. It is connected by two prefabricated shear wall units through the way of up-and-down splicing. And the shear wall structure unit consists of steel plates on both sides with concrete filled in them. In the edge, set concrete-filled steel tubular columns as the edge constraint members. The steel plates and the concrete in it are connected by M10 bolts to make the steel plate and the concrete work together. In the upper part of the unit, the reinforced flitch plates are welded, and in the lower part, the strengthened connection plates are welded. To draw an analogy, the reinforced flitch plates are like a "socket", and the lower strengthened connection plates are like a "plug", which are convenient for the upper and lower units to be connected with each other. This type of shear wall is called "socket-plug" type composite shear wall (SPRC). The upper and lower units are fixed by M24 high-strength bolts with the horizontal connection height set at about 1.3 m storey height. A gap will be set at the junction, and later concrete will be poured into the gap to connect the upper and lower units as a whole, and the design drawing of the composite shear wall is shown in Figure 1.

In the process of construction, insert the upper unit into the lower unit to the position of the limit block. Then link the upper and lower units together with the M24 high-strength bolts. Afterward, fill up the space created by the limit block with on-site concrete pouring to complete the construction of the horizontal connection, as shown in Figure 2.

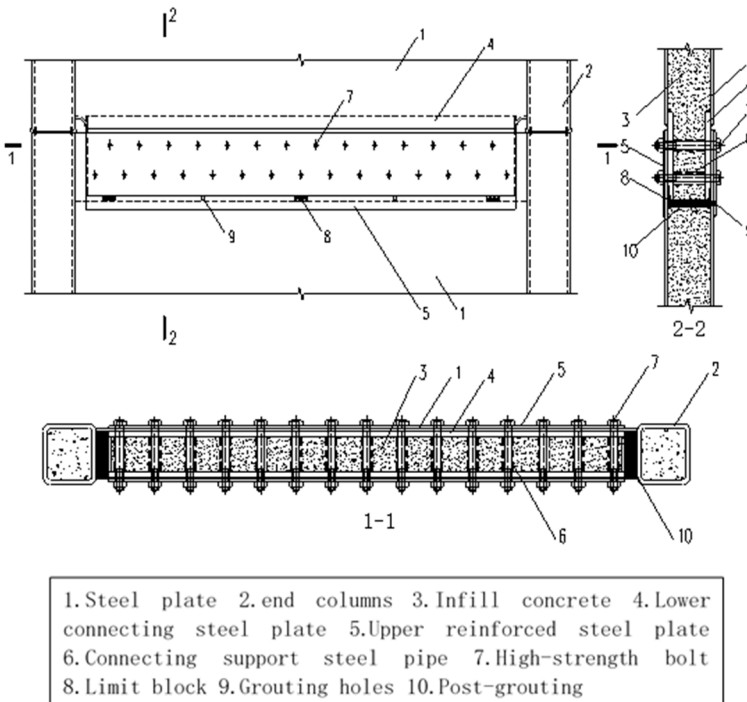

**Figure 1.** The constitution diagram of composite shear wall.

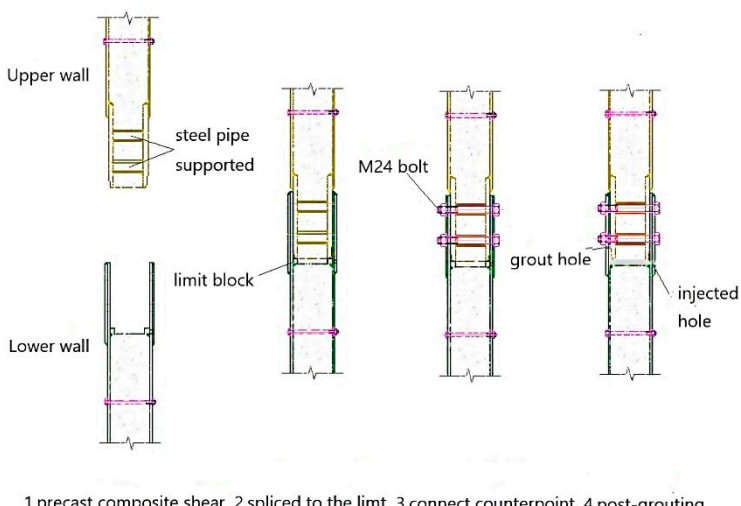

**Figure 2.** Connection construction method of the upper and lower composite shear walls.

## 2.2. Modeling of Composite Shear Wall

The steel plate-concrete composite shear wall connected by bolts consists of steel plates on both sides, edge constraint steel tube column, in-filled concrete, connecting steel plate in the lower part of the shear wall, reinforced flitch plate in the upper part of the shear wall, high-strengthen bolts and bolt sleeve. The size of each member is shown in Table 1, and the model explanation of each member in the upper and lower shear wall units is shown in Figures 3 and 4.

**Table 1.** Dimension information of members.

| Member | Dimension Information |
| --- | --- |
| Steel plates on both sides | 1950 mm × 1500 mm × 6 mm steel plate |
| Edge constraint steel tube column | 200 mm × 3 mm square steel tube column |
| Lower connecting steel plate | 1870 mm × 350 mm × 10 mm steel plate |
| Upper reinforced flitch plate | 1870 mm × 350 mm × 10 mm steel plate |
| Bolts | M24 high-strengthen bolts |

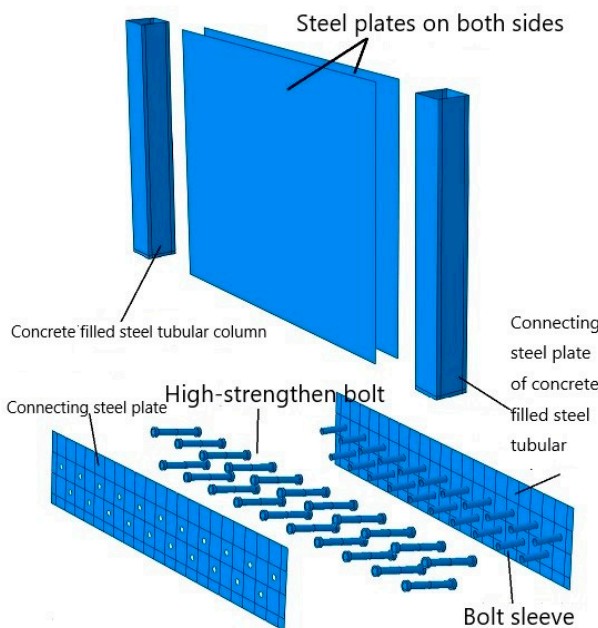

**Figure 3.** Description of components of upper shear wall unit.

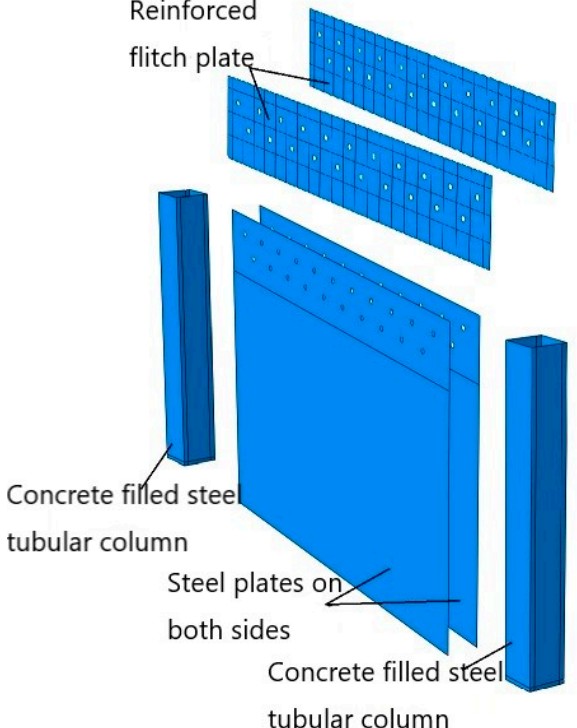

**Figure 4.** Description of components of lower shear wall unit.

Fully considering the contact problems that might be encountered in the actual situation, the simulation and setup are carried out carefully in the process of assembly and installation, including the contact between the bolt head and the steel plate, the hard touching between the shank and the inner hole wall, the contact between the lateral steel plate and the connecting steel plate, the contact between the lateral steel plate and the reinforced flitch plate, the contact between the steel plate and the infilled concrete, the contact between the bolt sleeve and the steel plate. The contact between the bolt head (nut) and the connecting plate is established by the penalty function, and the friction coefficient is 0.4. As to the contact between the nut and the steel plate, the steel plate surface is selected as the main surface and the inner side of the nut as the second surface. And the friction coefficient of 0.4 is set in the tangential behavior when setting the role attribute and the normal behavior is set to hard contact, to better simulate the contact between the nut and the plate surface with friction. The steel plate wall, the connecting steel plate, and the reinforced flitch plates are connected by welding. In the finite element model, the TIE constraint is used to simulate the welding connection, which can limit 6 degrees of freedom. The upper and lower shear wall units form the shape of plugs and sockets. The assembly diagrammatic drawing of the software simulation is shown in Figure 5.

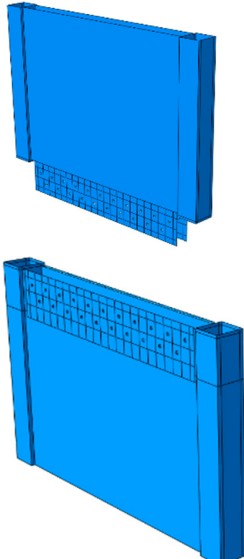

**Figure 5.** Diagrammatic drawing of simulation assembly 1.

In setting the boundary conditions, the constraint conditions are simulated based on the shear wall experimental study of the Ma [17]. The concrete section and the steel section at the bottom of the shear wall are coupled to one point, and the three translational and three rotational directions of freedom at this point are constrained so that the bottom surface is completely fixed. In addition, the concrete and steel sections at the top surface are coupled to one point, and the degrees of freedom in the URx, URy, and URz directions are constrained at this point to simulate the axial pressure in the vertical direction and the shear force in the horizontal direction. The boundary conditions of finite elements are set as shown in Figure 6.

When establishing the numerical model, the grid division of finite elements is firstly carried out. Mesh generation requires seed setting, mesh control and unit type selection, then its model can mesh. Like the seed set, the mesh generation also differs between independent entities and non-independent entities. For the seed setting in the model, the influence of the bolt spacing needs to be considered in the parametric analysis, and the final meshing settings of its model components are shown in Table 2. Change the relevant parameters when using the ABAQUS to make finite element analysis. Then 11 different finite element models were established, among which SPRC-1 to SPRC-4 are different axial compression ratio models, SPRC-5 to SPRC-7 are different aspect ratio models, SPRC-8

to SPRC-10 are models whose end column sizes are different. DSCW-1 is a conventional double steel plate composite shear wall, which has the same dimensions as SPRC-2, but without bolt connection. The relevant information of all models is shown in Table 3.

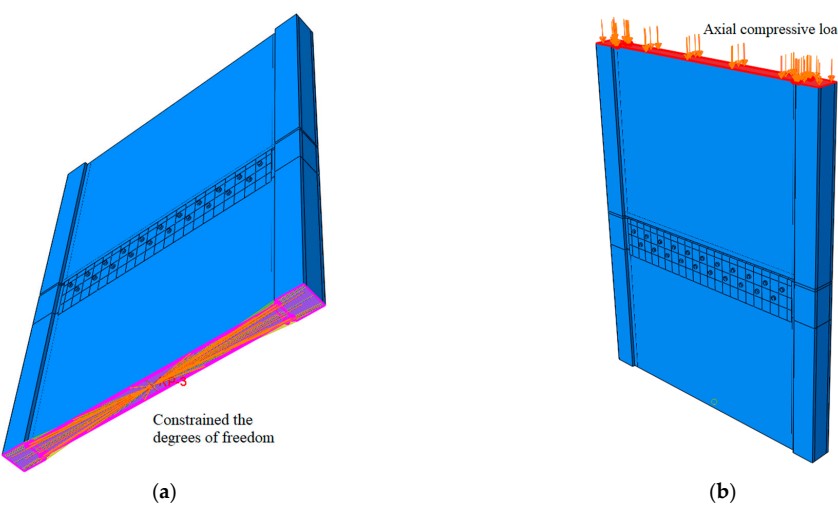

(**a**)  (**b**)

**Figure 6.** Diagrammatic drawing simulation of boundary conditions. (**a**) Bottom simulation completely fixed. (**b**) Top simulation applies axial and shear forces.

**Table 2.** Grid cell types of model components.

| Component | Mesh Shape | Delineation Technique | Unit Type |
|---|---|---|---|
| Bolt | Wedge | Scan | C3D6 |
| Post pouring concrete | Hexahedron | Structure | C3D8 |
| End steel tube column | Quadrilateral | Structure | S4 |
| Steel plate wall | Quadrilateral, triangle | Structure, freedom | S4, S3 |

**Table 3.** Relevant parameters of each model.

| Number | Axial Compression Ratios | Component Height (mm) | End Column Size (mm × mm) |
|---|---|---|---|
| SPRC-1 | 0.4 | 3000 | 200 × 200 |
| SPRC-2 | 0.5 | 3000 | 200 × 200 |
| SPRC-3 | 0.6 | 3000 | 200 × 200 |
| SPRC-4 | 0.7 | 3000 | 200 × 200 |
| SPRC-5 | 0.5 | 2700 | 200 × 200 |
| SPRC-6 | 0.5 | 3300 | 200 × 200 |
| SPRC-7 | 0.5 | 3600 | 200 × 200 |
| SPRC-8 | 0.5 | 3000 | 180 × 180 |
| SPRC-9 | 0.5 | 3000 | 220 × 220 |
| SPRC-10 | 0.5 | 3000 | 240 × 240 |
| DSCW-1 | 0.5 | 3000 | 200 × 200 |

### 2.3. Material Properties

Considering that the plasticity of steel is an important factor affecting the overall performance of the structure, the three-fold line of the stress-strain relationship is adopted in the steel plate. The three-fold curve is relatively complex, but the analysis and description of large deformation can be more accurate, and its stress-strain relationship is shown in Figure 7a. In the process of tension and compression, concrete will cause the plastic development and rigidity of the material to decrease, therefore the mechanical properties will become relatively complex. The concrete damage plastic (CDP) is used to simulate the mechanical behavior of the concrete. The stress-strain curves under tension and compression are shown in Figure 7b.

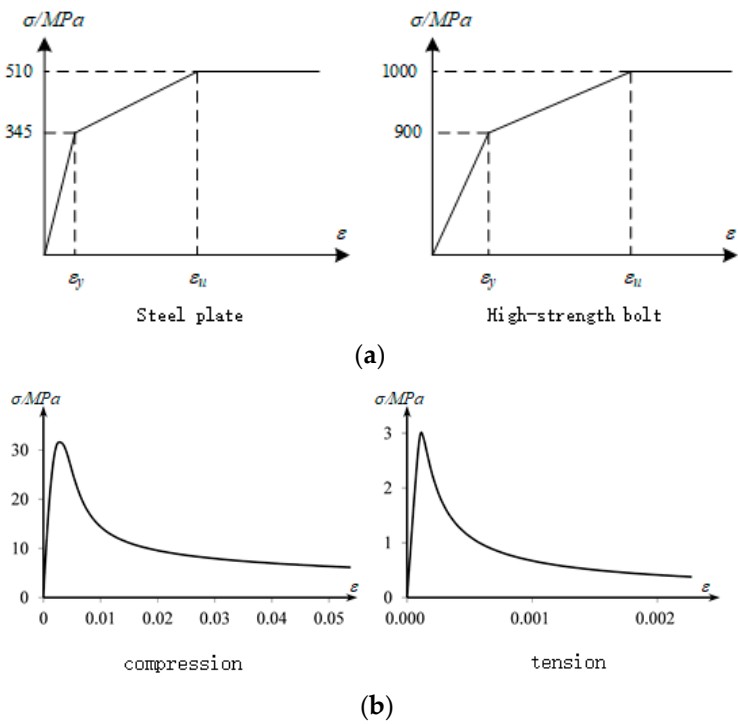

**Figure 7.** Stress-strain curves of steel and concrete. (**a**) Stress-strain curves of steel. (**b**) Stress-strain curves of concrete.

## 3. Analysis of Mechanical Performance of Composite Shear Walls

### 3.1. Comparison between Assembled and Conventional Composite Shear Walls

The assembled composite shear walls are bolted together, which makes the construction more convenient and faster and also allows the walls to achieve the load-bearing capacity of conventional composite shear walls. Therefore, monotonic loads were applied to the SPRC-2 specimen and DSCW-1 specimen respectively to compare the stress performance and damage forms of the two models. The displacement-load curves of their two models, as shown in Figure 8, show that the load-displacement curves of both are not much different and the initial stiffness is almost the same. While the ultimate bearing capacity of the DSCW-1 model reaches 3018.2 kN, which is 5.6% higher than one of the SPRC-2 model of 2859.2 kN. Thus, it can be considered that the shear wall, which connects through bolts, can achieve the load-bearing capacity of a conventional double steel plate the composite shear wall.

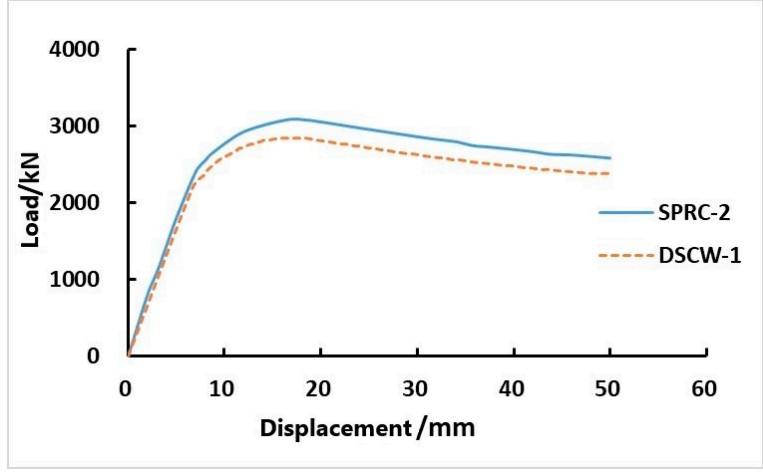

**Figure 8.** Diagrammatic drawing of simulation assembly 2.

Comparing the damage forms of the two specimens, the equivalent plastic strain (PEEQ) of concrete and steel members of the SPRC-2 wall at the completion of load application are shown in Figures 9 and 10. The crushed area on the compressed side of the concrete expands to the center of the wall. Most of the equivalent plastic strain values in the lower part of the compressed side of the concrete wall reach 0.012 or more, namely, half of the concrete in the bottom of the wall is severely damaged by compressive damage and has basically lost its bearing capacity. The compressed side of the steel plate has already shown very obvious out-of-plane flexural damage and the maximum of equivalent plastic strain in the compressed area reaches 0.0465, while most of the steel plate concrete in the middle connection part is still in elastic behavior state. Finally, its damage is in the form of concrete crushing at the bottom of the compressed side and flexural damage in the bottom of the steel plate.

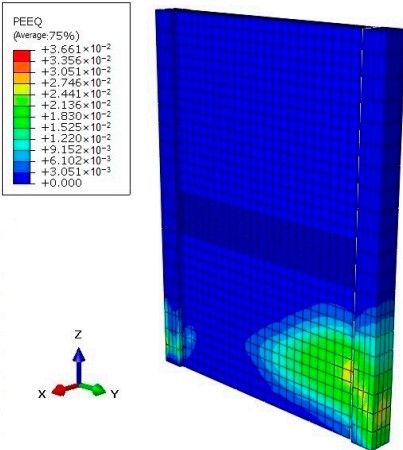

**Figure 9.** PEEQ of steel members for SPRC-2 (1).

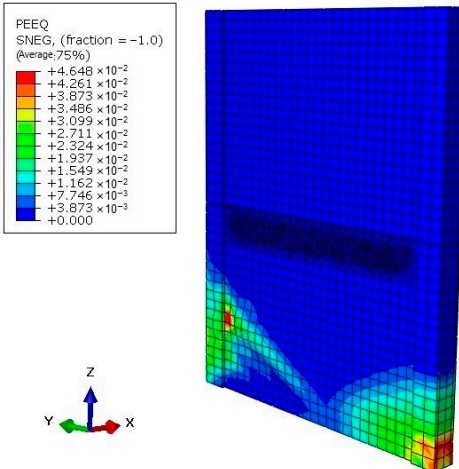

**Figure 10.** PEEQ of concrete for SPRC-2 (1).

Analyzing the damage form of the DSCW-1 model at the final moment, the equivalent plastic strain cloud charts of steel members and concrete are shown in Figures 11 and 12. What can be seen from the figures is that the steel members of the model appeared out-of-plane drumming at the bottom of the wall, and the concrete at the bottom also showed obvious crushing damage, which is basically the same as the damage form of the SPRC-2 specimen. Meanwhile, the initial stiffness and bearing capacity of the two specimens were analyzed comprehensively, and it can be considered that the assembled composite shear wall can reach the mechanical performance level of the conventional composite shear wall.

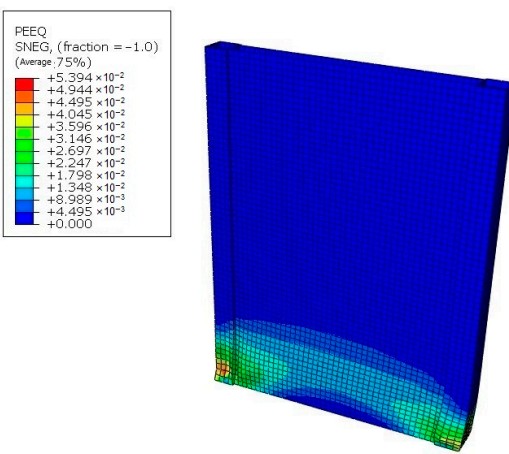

**Figure 11.** PEEQ of steel members for DSCW-1.

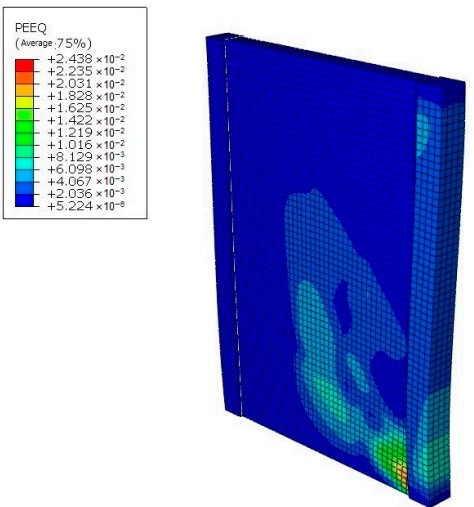

**Figure 12.** PEEQ of concrete for DSCW-1.

### 3.2. Analysis of Hysteresis Performance of Composite Shear Walls

The ductility, energy dissipation capacity and hysteresis performance of the composite shear wall were analyzed by applying reciprocating load to the composite shear wall structure and observing the hysteresis curve, skeleton curve, energy dissipation coefficient and other related indexes. Reciprocating load is applied to the composite wall, and the loading method is referred to as the loading system as shown in Figure 13. The reciprocating load was loaded with 2 mm increments per cycle. And each load level was cycled only once, while the maximum displacement of the load was selected as 30 mm.

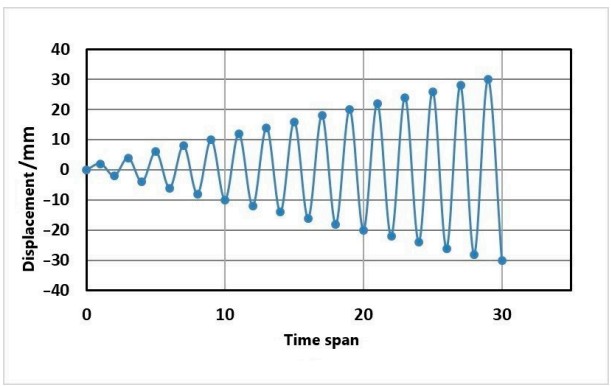

**Figure 13.** Cyclic loading system.

Select the specimen SPRC-2 as a typical one. According to the related cloud chart from software simulation, analyze the damaged form of the composite shear wall. At the end of loading, the equivalent plastic strain of the steel structure and concrete structure is shown in Figures 14 and 15. It can be seen that the final damage form of the composite wall is that the steel pipe columns on both sides and the basic angle of the steel plate are seriously buckled with the lower side of the concrete is almost completely crushed.

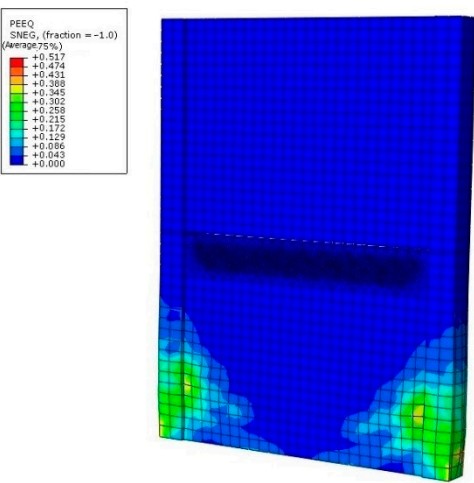

**Figure 14.** PEEQ of steel members for SPRC-2 (2).

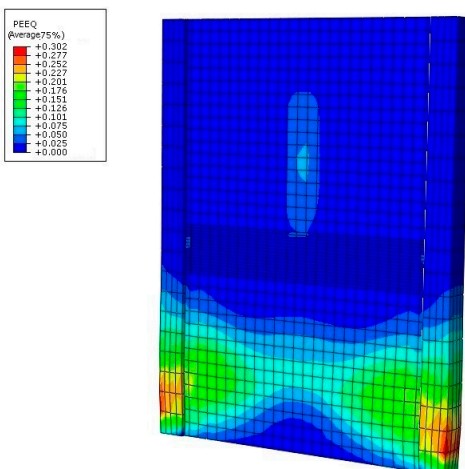

**Figure 15.** PEEQ of concrete for SPRC-2 (2).

### 3.2.1. Analysis of Hysteretic Curve

The hysteresis curve is the load-deformation curve of the structure under cyclic loading. It can reflect the deformation capacity, stiffness degradation and energy dissipation of shear walls under cyclic loading, which are the basis for studying restoring force models and nonlinear earthquake response analysis. It is also known as restoring force curve. The hysteresis curves of all composite shear wall specimens under different parameters are shown in Figure 16. Because the concrete damage plasticity (CDP) exhibit softening behavior and stiffness degradation of the material model, the singularity of the stiffness matrix in the implicit analysis procedure leads to convergence difficulties, as models such as SPRC-1, SPRC-3, etc. fail to calculate up to a predetermined maximum 30 mm displacement.

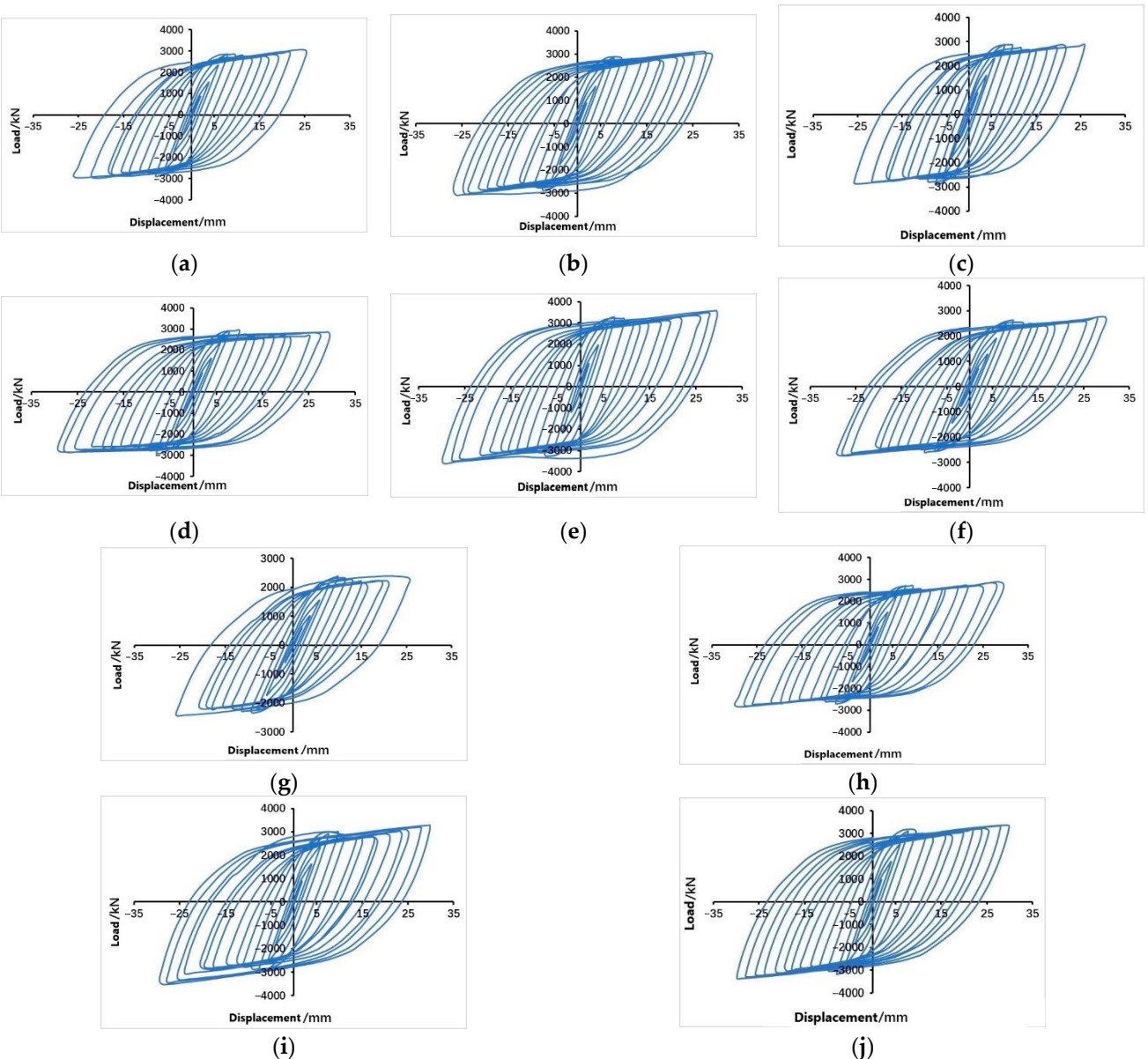

**Figure 16.** The hysteresis curves of different parameter models. (**a**) SPRC-1. (**b**) SPRC-2. (**c**) SPRC-3. (**d**) SPRC-4. (**e**) SPRC-5. (**f**) SPRC-6. (**g**) SPRC-7. (**h**) SPRC-8. (**i**) SPRC-9. (**j**) SPRC-10.

From the analysis of the hysteresis curves of all shear walls in Figure 16, it can be seen that the shear walls have the following rules during the loading process.

(1) The hysteresis curves at the initial stage are approximately linear, and in the elastic working stage, with almost no residual deformation.

(2) With the reciprocal loads continuing to be applied, the hysteresis curves of the shear wall begin to be spindle-shaped and relatively full. The slope of the curve gradually decreases, which indicates that the stiffness of the shear wall begins to reduce; the unloading point of the hysteresis loop begins to be enriched with increasing residual deformation, indicating that the shear wall has developed into the elastic-plastic stage.

(3) Continuing to apply reciprocal loads after reaching the peak load, the slope of the hysteresis curve decreases faster for shear walls with different parameter conditions, namely, the stiffness degeneration of the shear wall develops rapidly and the residual deformation at the unloading point of the hysteresis curves increases.

Comparing the shear walls under different axial compression ratios, it is found that the effects of axial pressure ratios on the hysteresis curves are very small. And there is

almost no difference in the fullness of the hysteresis curve of each shear wall, with all being relatively full. Analyzing the hysteresis performance under different aspect ratios and end column sizes, it can be seen that only the hysteresis curve of the specimen with a height of 3600 mm owns a smaller bounded area, and the fullness of the hysteresis curves under other height conditions is similar. The best hysteresis performance appears at a height of about 3000 mm, and the height should not be too high. For the end column size, it has less influence on the hysteresis performance, and the energy dissipation capacity of each specimen is relatively close.

### 3.2.2. Analysis of Skeleton Curve

The envelope line obtained by connecting the maximum point of each lap of the loading curve of the hysteresis curve is the skeleton curve. The skeleton curve is an important index in the study of nonlinear seismic loading, and the maximum points cannot be crossed out of the skeleton curve during the whole process of loading. The skeleton curve can reflect the strength, stiffness and ductility of the shear wall.

Figure 17 gives a comparison diagram of the skeleton curves of each shear wall under different axial compression ratios, height-width ratios and end column sizes. It shows that at the beginning of loading, the load and displacement grow approximately linearly, so they are in an elastic working condition. After that, it develops into nonlinear working conditions, the skeleton curves appear obvious turns, the stiffness of the wall starts to reduce, and the shear wall develops into the yield failure stage and exhibits certain ductility.

Compared with the shear wall under different axial compression ratios, there are no big differences in the skeleton curve. It can be seen that the axial compression ratio has little effect on the hysteretic performance of the shear wall. Comparing the shear walls with different aspect ratios, as the aspect ratio increases, the walls' initial stiffness and peak value of horizontal load decrease. The reason is that the increase in the aspect ratio augments the bending moment of the shear wall during bending. Comparing the shear walls with different end column sizes, the peak load of the shear wall increases with the increase of the end column size, while the initial stiffness has no obvious difference.

### 3.2.3. Analysis of Ductility

The calculation of ductility is usually expressed by the ratio of ultimate displacement to yield displacement. The equivalent yield displacement is obtained using the R. Park method. Take 75% of the peak load point of the member, connect this point with O point, extend the line and the horizontal line over M to intersect at A point, make a vertical line over A point to get the yield point Y, as shown in Figure 18. The ultimate displacement is chosen to be the displacement point corresponding to when the peak load drops to 85%.

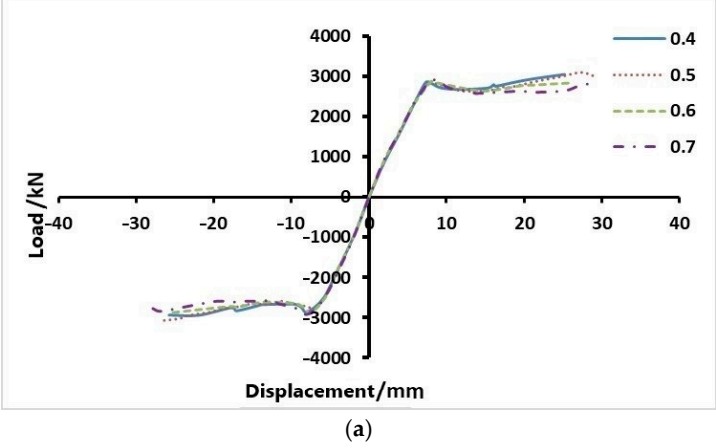

(**a**)

**Figure 17.** *Cont.*

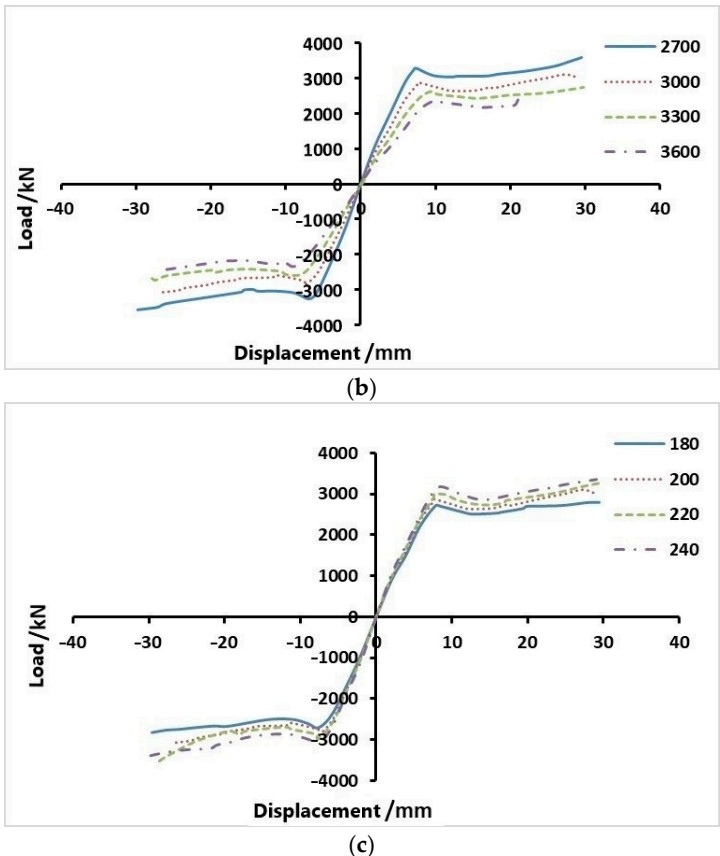

**Figure 17.** Skeleton curves under different parameters. (**a**) Different axial compression ratios. (**b**) Different height-width ratios. (**c**) Different sizes of end column.

From Table 4, it can be seen that when the axial compression ratio increases, the yield bearing capacity of the shear wall decreases. The maximum ductility of the structure is about 3.74 when the axial compression ratio is 0.5. By increasing the aspect ratio of the composite shear wall, the yield load and peak load of the structure decrease and the ductility tends to decrease. By increasing the size of the end columns, the yield load and peak load of the structure increase, but the ductility of the structure decreases.

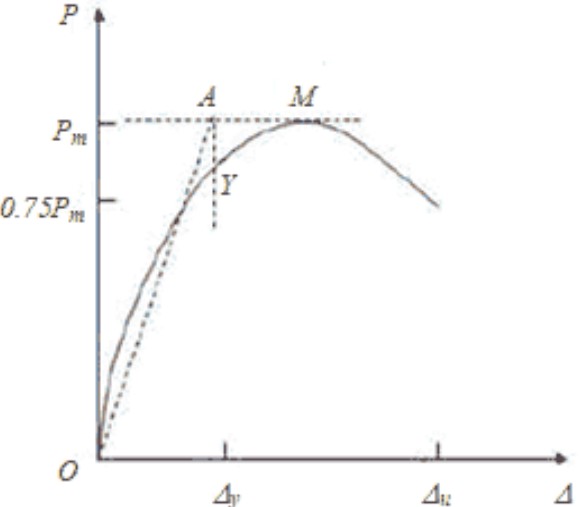

**Figure 18.** Ductility by secant stiffness method.

**Table 4.** Results for ductility.

| Number | Yield Load/kN | Yield Displacement/mm | Peak Load/kN | Peak Displacement/mm | Ultimate Displacement/mm | Ductility |
|--------|---------------|----------------------|--------------|---------------------|-------------------------|-----------|
| SPRC-1 | 2791 | 7.72 | 3052 | 25.38 | 25.38 | 3.28 |
| SPRC-2 | 2768 | 7.72 | 3088 | 28.93 | 28.93 | 3.74 |
| SPRC-3 | 2708 | 7.19 | 2846 | 25.56 | 25.56 | 3.55 |
| SPRC-4 | 2622 | 7.81 | 2792 | 28.07 | 28.07 | 3.58 |
| SPRC-5 | 3220 | 7.66 | 3548 | 29.73 | 29.73 | 3.88 |
| SPRC-6 | 2462 | 8.56 | 2711 | 29.72 | 29.72 | 3.47 |
| SPRC-7 | 2250 | 9.35 | 2365 | 21.61 | 21.61 | 2.31 |
| SPRC-8 | 2638 | 7.39 | 2796 | 29.22 | 29.22 | 3.95 |
| SPRC-9 | 2968 | 7.92 | 3264 | 29.22 | 29.22 | 3.68 |
| SPRC-10 | 3158 | 8.15 | 3363 | 29.06 | 29.06 | 3.56 |

## 4. Conclusions

(1) Comparing the prefabricated composite shear wall with the conventional composite shear wall of the same size, their damage forms are basically the same, and their initial stiffness and ultimate bearing capacity do not have much difference. The newly proposed composite shear wall can be prefabricated in the factory and bolted on site to speed up the installation process.

(2) The plug-in composite shear wall shows great hysteretic performance and high bearing capacity under the reciprocating load. The hysteretic curves of this composite shear walls are full under different parameters, which indicates that the composite shear walls have strong energy dissipation capacity and good seismic performance. Therefore, it can be widely applied in the field of prefabricated steel structure construction.

(3) The effects of height-span (aspect) ratio, axial compression ratio and end column size on the hysteretic behavior of composite shear walls are studied. The aspect ratio of the composite shear wall has a great influence on the seismic performance of the structure. The hysteretic performance of the shear wall with different aspect ratio is obviously different, and the hysteretic performance of the structure is the best when the aspect ratio is 1.5. With the increase of aspect ratio, the bearing capacity and ductility of joints decrease. The axial compression ratio shows better seismic performance when the axial compression ratio is 0.3 to 0.7. Moreover, increasing the size of the end column can improve the bearing capacity of the structure, but the ductility will decrease.

**Author Contributions:** Conceptualization, Z.W. and Z.M.; methodology, X.S.; software, X.S., Z.J. and P.L.; validation, X.S., Z.J. and Y.Y.; formal analysis, X.S. and P.L.; investigation, Y.Y. and Z.W.; resources, P.L.; data curation, P.L.; writing—original draft preparation, X.S. and P.L.; writing—review and editing, Z.J. and Y.Y.; visualization, P.L. and Y.Y.; supervision, Z.M.; project administration, Z.M.; funding acquisition, Z.M. All authors have read and agreed to the published version of the manuscript.

**Funding:** This research was funded by the National Natural Science Foundation of China grant number 51578064 and the Natural Science Foundation of Beijing Municipality grant number 8172031).

**Data Availability Statement:** All the data supporting the results were provided within the article.

**Conflicts of Interest:** The authors declare no conflict of interest.

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
