# Peer review of "Analysis of Design Method and Mechanical Properties of Plug-In Composite Shear Wall"

_metals, doi:10.3390/met13010177_

Round 1
Reviewer 1 Report
1. Please explain the reason for the different displacement between hysteresis curves in fig 15.
2. PEEQ of concrete is quite different compared to others, as shown in fig 11, is there any specific reason for having a different distribution of loads on the wall, justify.
3. In fig 12, what is the dwell time during the transmission of displacement for Fig 12.
4. Is it recommended to have the column size 200mm, irrespective of the total size of the unit, or may it vary proportionally? Please provide detailed information.
5. Please provide a quantitative analysis of the ductility for all the members.
6. Results and discussion section must be improved by adding more scientific discussions.
Author Response
Response to Reviewer 1 Comments
Point 1: Please explain the reason for the different displacement between hysteresis curves in fig 15.
Response 1: The expected maximum displacement of loading for each model is 30mm. Due to the serious damage or deformation of concrete material in the model at the late stage of loading, the model calculation does not converge, thus some models fail to reach the predetermined maximum displacement of 30mm, and thus the calculation is terminated.
Point 2: PEEQ of concrete is quite different compared to others, as shown in fig 11, is there any specific reason for having a different distribution of loads on the wall, justify.
Response 2: Figure 11 shows the PEEQ values of the concrete at the final moment of DSCW-1. It can be seen from the model deformation that this model has an out-of-plane buckling at the bottom of the wall, and its final damage is different from the other models, so the PEEQ distribution of DSCW-1 is also different.
Point 3: In fig 12, what is the dwell time during the transmission of displacement for Fig 12.
Response 3: The loading rate of the general proposed static test is very low, so the stress and strain changes caused by the loading rate have little effect on the test results and can be neglected. In the finite element model, the static analysis method is used without considering the dynamic related influence factors, and it is not necessary to set the cyclic loading time.
Point 4: Is it recommended to have the column size 200mm, irrespective of the total size of the unit, or may it vary proportionally? Please provide detailed information.
Response 4: The hysteresis performance of the combined walls with different aspect ratios, axial compression ratios and end column sizes were analyzed. From the results, the aspect ratio has an obvious influence on the hysteresis performance of the compsite shear wall structures, and the hysteresis curve is fuller when the height is around 3000 mm. When the axial pressure ratio was 0.4-0.7, the shear walls all showed good performance. The structural bearing capacity increases slightly with the increase of end column size when the column size is from 180 to 240. Therefore, giving the recommended size under common height, the end column size can be 200mm, which is economical and also has good performance requirements. When the total size of the composite wall is changed, the size of the end columns can be adjusted appropriately to meet the performance requirements.
Point 5: Please provide a quantitative analysis of the ductility for all the members.
Response 5: A new section 2.2.3 has been added for the ductility analysis of the composite shear wall.
Point 6: Results and discussion section must be improved by adding more scientific discussions.
Response 6: Revisions were made to the conclusions and content of the manuscript.

Reviewer 2 Report
The manuscript entitled "Analysis of design method and mechanical properties of plug-in composite shear wall” presented a newly developed assembly of shear walls.
This manuscript lacks clarity and is poorly organized. This reviewer cannot understand if the manuscript is modeling or experimental work. This manuscript cannot be accepted and should be rejected.
Technical comments:
1- The authors should provide the number lines to easily track the review.
2- The literature should be expanded to highlight how their study is providing a different approach or adding significantly to what has been done and include the most related studies to highlight the new approach developed in this manuscript.
3- Figure 1: The provided figure is not enough to demonstrate the details in section 1.1. This reviewer highly recommends providing a new three-dimensional figure showing these details or at least other side and plan views.
4- Figure 2: The quality of the provided figure is very poor.
5- Section 1.2: The information was repeated in section 1.2. Why?
6- Simulations of the mechanical properties of materials are missing in the provided modeling. What were the constitutive models for concrete and steel? Did the authors consider the material and geometry nonlinearities?
7- Section 1.2: The authors mentioned "The steel plate wall, the connecting steel plate, and the reinforced flitch plates are connected by welding during the modeling and assembly". What was the contact between the welding and these parts?
8- Section 2.1: Where are the details of specimens SPRC-2 and DSCW-1? The authors did not provide any information about these specimens before. Where is the verification of the developed modeling?
9- Figure 7: This figure is not showing any verification for the developed model. Why did the authors try to prove the effectiveness of the new shear wall? If so, only one comparison can confirm this?!
10- No information was provided about how did the authors model the damage in the developed shear walls.
Author Response
Response to Reviewer 2 Comments
Point 1: The authors should provide the number lines to easily track the review.
Response 1: Line numbers have been added to the right side of the manuscript.
Point 2: The literature should be expanded to highlight how their study is providing a different approach or adding significantly to what has been done and include the most related studies to highlight the new approach developed in this manuscript
Response 2: Some references have been added to this manuscript. Revised some paragraphs to better highlight the new structural form of this thesis.
Point 3: Figure 1: The provided figure is not enough to demonstrate the details in section 1.1. This reviewer highly recommends providing a new three-dimensional figure showing these details or at least other side and plan views.
Response 3: New diagrams and cross-sections are used to show the newly proposed composite shear wall in Fig.1.
Point 4: Figure 2: The quality of the provided figure is very poor.
Response 4: The image quality of Figure 2 was adjusted.
Point 5: Section 1.2: The information was repeated in section 1.2. Why?
Response 5: Section 1.2 shows the model built in finite elements, including members such as end columns, double-layered steel plates, in-fill concrete, and high-strengh bolts.
Point 6: Simulations of the mechanical properties of materials are missing in the provided modeling. What were the constitutive models for concrete and steel? Did the authors consider the material and geometry nonlinearities?
Response 6: Section 1.3 was added to supplement the streee-strain relationships of steel and concrete. The steel is considered plastic and a three-fold curves is used. The concrete plastic damage (CDP) was used for infill concrete.
Point 7: Section 1.2: The authors mentioned "The steel plate wall, the connecting steel plate, and the reinforced flitch plates are connected by welding during the modeling and assembly". What was the contact between the welding and these parts?
Response 7: In the actual structure, the steel plate walls, connecting plates and reinforced plates are connected to each other by welding. In the finite element model, the TIE constraint is used to simulate the welding connection. TIE constraint can limit 3 rotational degrees of freedom and 3 translational degrees of freedom, which can well simulate the constraint effect of welding.
Point 8: Section 2.1: Where are the details of specimens SPRC-2 and DSCW-1? The authors did not provide any information about these specimens before. Where is the verification of the developed modeling?
Response 8: The dimensions and structural forms of the relevant models are presented in Section 1.2. “socket-plug” type composite shear wall (SPRC) is a new type composite shear wall proposed in this paper, with different dimensions named by SPRC1-10. And DSCW is a conventional double steel plate-concrete composite shear wall as a contral model.
Additional descriptions are given in the manuscript.
Point 9: Figure 7: This figure is not showing any verification for the developed model. Why did the authors try to prove the effectiveness of the new shear wall? If so, only one comparison can confirm this?!
Response 9: By comparing the load-displacement curves of the “socket-plug” type composite shear wall (SPRC) and the conventional double steel plate combination shear (DSCW), it can be seen that the newly proposed composite shear wall has similar stiffness and bearing capacity as the conventional composite shear wall. It is shown that the “socket-plug” connection method does not affect the performance of the composite shear wall, and it facilitates the use of assembled installation method for the new type composite shear wall, which is convenient for the standardized prefabrication of the members and accelerates the installation speed.
Point 10: No information was provided about how did the authors model the damage in the developed shear walls.
Response 10: Elastic-plastic constitutive models was used for steel and concrete plastic damage (CPD) was used for concrete’s constitutive models. In the finite element results, the equivalent plastic strain (PEEQ) was used to predict the plastic accumulation occurring in the structure, from which the damage location of the structure was determined.

Round 2
Reviewer 1 Report
The quality of the Revised manuscript is improved and it can be considered for publication.
Reviewer 2 Report
The authors have addressed most of the reviewer's comments.